# Origin, Genetic Variation and Molecular Epidemiology of SARS-CoV-2 Strains Circulating in Sardinia (Italy) during the First and Second COVID-19 Epidemic Waves

**DOI:** 10.3390/v15020277

**Published:** 2023-01-18

**Authors:** Angela Maria Rocchigiani, Luca Ferretti, Alice Ledda, Antonello Di Nardo, Matteo Floris, Piero Bonelli, Federica Loi, Maria Laura Idda, Pier Paolo Angioi, Susanna Zinellu, Mariangela Stefania Fiori, Roberto Bechere, Paola Capitta, Annamaria Coccollone, Elisabetta Coradduzza, Maria Antonietta Dettori, Maria Caterina Fattaccio, Elena Gallisai, Caterina Maestrale, Daniela Manunta, Aureliana Pedditzi, Ivana Piredda, Bruna Palmas, Sara Salza, Anna Maria Sechi, Barbara Tanda, Maria Paola Madrau, Maria Luisa Sanna, Simonetta Cherchi, Nicoletta Ponti, Giovanna Masala, Roberto Sirica, Eloisa Evangelista, Annalisa Oggiano, Giantonella Puggioni, Ciriaco Ligios, Silvia Dei Giudici

**Affiliations:** 1Istituto Zooprofilattico Sperimentale della Sardegna, 07100 Sassari, Italy; 2Pandemic Sciences Institute and Big Data Institute, Li Ka Shing Centre for Health Information and Discovery, Nuffield Department for Medicine, University of Oxford, Oxford OX1 2JD, UK; 3UK Health Security Agency, Colindale, London NW9 5EQ, UK; 4The Pirbright Institute, Ash Road, Pirbright, Woking GU24 0NF, UK; 5Department of Biomedical Sciences, University of Sassari, 07100 Sassari, Italy; 6Osservatorio Epidemiologico Veterinario Regionale, Istituto Zooprofilattico Sperimentale della Sardegna, 09125 Cagliari, Italy; 7Institute for Genetic and Biomedical Research (IRGB), National Research Council (CNR), 07100 Sassari, Italy; 8Ames Polydiagnostic Group Center SRL, 80013 Napoli, Italy

**Keywords:** SARS-CoV-2, spike (S) protein, whole genome sequencing, molecular epidemiology, genetic diversity

## Abstract

Understanding how geography and human mobility shape the patterns and spread of infectious diseases such as COVID-19 is key to control future epidemics. An interesting example is provided by the second wave of the COVID-19 epidemic in Europe, which was facilitated by the intense movement of tourists around the Mediterranean coast in summer 2020. The Italian island of Sardinia is a major tourist destination and is widely believed to be the origin of the second Italian wave. In this study, we characterize the genetic variation among SARS-CoV-2 strains circulating in northern Sardinia during the first and second Italian waves using both Illumina and Oxford Nanopore Technologies Next Generation Sequencing methods. Most viruses were placed into a single clade, implying that despite substantial virus inflow, most outbreaks did not spread widely. The second epidemic wave on the island was actually driven by local transmission of a single B.1.177 subclade. Phylogeographic analyses further suggest that those viral strains circulating on the island were not a relevant source for the second epidemic wave in Italy. This result, however, does not rule out the possibility of intense mixing and transmission of the virus among tourists as a major contributor to the second Italian wave.

## 1. Introduction

Emerging infectious diseases (EIDs) are infections that have appeared in a population for the first time, or that are already present, but rapidly spreading with increasing incidence [1]. The majority of EIDs (~60%) originate from animal reservoirs, and approximately one in five are caused by vector-borne pathogens [2]. The origins of zoonotic EIDs are related to several environmental and sociocultural variables that are deeply interconnected. In the last decades, as a direct consequence of globalization, we have witnessed a dramatic rise in urbanization and a significant destruction of natural habitats which, likely by altering vector ecologies, affect the distribution and diffusion of infectious diseases [3,4]. As underlined by the World Health Organization (WHO), the emergence and spread of EIDs represent a threat to global health, which is difficult to predict and accompanied by alarming socio-economic impacts [1,5]. The severe acute respiratory syndrome coronavirus 2 (COVID-19) is the most recent EID, which is still raising great concern worldwide despite our improved knowledge of its transmission patterns and the ongoing vaccination campaigns [6]. COVID-19 is caused by the Coronavirus SARS-CoV-2, which is part of a large family of enveloped RNA viruses able to infect both animals and humans. In humans, coronaviruses are usually responsible for mild to moderate upper-respiratory tract illnesses. Whether SARS-CoV-2 shares the same zoonotic origin as SARS-CoV [7] is still debated, but it is certain that the COVID-19 pandemic has greatly outnumbered the number of cases of both SARS and MERS [8].

From its first occurrence in December 2019 in Wuhan (China), SARS-CoV-2 has dramatically spread worldwide, becoming a major global threat to public health following the declaration of its pandemic status by the WHO in March 2020 [1]. To date, more than half a billion cases and millions of deaths caused by different variants of concern (VOCs) have been confirmed by the WHO [9]. The first and second epidemic waves, caused by the wild-type and novel variants (B.1.177 and Alpha, respectively), were characterized by a high mortality rate, owing to high virus pathogenicity in a highly susceptible population and the unpreparedness of the global health system [10]. Subsequently, the combined availability of licensed vaccines, new therapeutic opportunities and non-pharmaceutical interventions drastically reduced the mortality. The continuous SARS-CoV-2 evolution with appearance of new VOCs (Beta, Gamma, Delta and Omicron) raises important concerns about the effectiveness of the current vaccination strategies [9,10,11].

Italy was one of the most affected countries in Europe during the first two pandemic waves, with a case-fatality rate of 2.67% of the total cases [12,13]. From the first confirmed case (24 February 2020) until the summer of 2021, about 4.3 million cumulative cases and 130,000 deaths were notified by the Italian Ministry of Health and reported to the WHO. After the first epidemic wave, which led to a national lockdown introduced on the 9th of March 2020 aimed at controlling the spread of the disease, the Italian epidemiological context has provided cause for alarm during the second wave (from September 2020 to March 2021), which was characterized by the appearance of more transmissible variants, first B.1.177 and then the Alpha VOC [11,14,15].

In the Italian epidemiological context (Figure 1A), the total number of new COVID-19 cases reported daily during the first wave was significantly less in comparison to the second wave (38,894 cases on 23 March, 2020; 242,062 cases on 9 November 2020), the total number of daily deaths (Figure 1B) was similar between the two waves (5339 deaths until 30 March 2020; 5151 deaths during the second wave until 30 November 2020) [9]. As reported by the Italian Integrated Surveillance System of Seasonal Influenza (InfluNET) [16] these data are strictly different from those reported for seasonal influenza. During the 2017–18 season, the peak of influenza was reached in the second week of 2018 with a level of incidence equal to 14.74 cases per 1000 patients. Considering a total of 1,309,903 patients who received assistance for 2017–2018 seasonal influenza, 19,308 cases were estimated. This number is about half of those reported during the first wave of COVID-19, and about 1% of the total new COVID-19 cases reported daily during the second wave [16].

Furthermore, the number of cases and deaths during the first wave was underestimated given the difference in the intensity of diagnostic testing between the two waves, as well as the presence of asymptomatic or mild cases, as has recently been demonstrated (Figure 2) [17,18].

In this paper, we focus on Sardinia, the second largest island in the Mediterranean Sea and one of the 20 Italian administrative regions. Sardinia is an important summer seaside holiday destination in Europe, characterized by hundreds of daily flights during the summer tourist season and few connections from autumn to spring. It is, therefore, not surprising that Sardinia was one of the last Italian regions infected during the first wave (March–May 2020), with one of the lowest COVID-19 cases incidences reported at that time (79.9 cases/100,000 inhabitants). Furthermore, its socio-geographic features—insularity, large rural areas, limited connectivity between major cities and low population density—have created a peculiar epidemiological scenario [13,17,19,20]. Tourism flow to the island during the summer season (June–September 2020) was restored after the lockdown measures, causing a sharp increase in the number of infections, and during the second wave Sardinia became the first Italian region categorized as high risk by the European Centre for Disease Prevention and Control (ECDC) [21]. After the easing of the restrictive measures in the summer season which caused the start of the second wave, more stringent measures were imposed by the decree of the President of the Council of Ministers (Dpcm) on 3 December [14,22] and considering that the Italian epidemiological context was still alarming, the whole country was subjected to strong lockdown measures aimed at controlling the spread of the disease. The entire nation’s territory was classified as red zone; thus, movement between regions and municipalities was forbidden, allowed only for work activities, health reasons, and urgent needs (i.e., to buy groceries, care for the elderly, or reach one’s own house) [13]. The aim of this work is to (i) investigate the genetic variation of SARS-CoV-2 strains circulating in Northern Sardinia during the first and second epidemic waves using Illumina and Oxford Nanopore Technologies (ONT) Next Generation Sequencing (NGS) approaches; and (ii) to clarify, through molecular epidemiology, the role that Sardinia played in the spread of COVID-19 infections across the Italian peninsula in autumn 2020.

## 2. Materials and Methods

### 2.1. Samples Collection

Nasopharyngeal swabs from individuals suspected to be infected with SARS-CoV-2 (n = 20,319) were collected from April 2020 to January 2021 during the period of COVID-19 surveillance implemented in Sardinia. FLOQSwabs were used for nasopharyngeal specimen collection, then placed into a transport tube with universal transport medium (UTM) (COPAN—Brescia, Italy) and shipped to the Istituto Zooprofilattico Sperimentale (IZS) della Sardegna in Sassari. No clinical information is available for the infected patients. All patient data were anonymized; only information about age, gender, and sampling date were collected.

### 2.2. Nucleic Acid Purification and Reverse Transcription Real-Time PCR

Viral RNA was purified from 200 µL of UTM by the MagMax Core Nucleic Acid Purification Kit (Applied Biosystems, Thermo Fisher Scientific, Waltham, MA, USA) in automated sample preparation workstation MagMAX Express 96 (Applied Biosystems), according to the manufacturer’s instructions.

Samples were tested by TaqPath™ COVID-19 CE-IVD RT-PCR Kit (Thermo Fisher) using the 7500HT Fast Real-Time PCR System (Applied Biosystems) according to manufacturer’s instructions. The kit is a multiplex assay that allows amplification of conserved regions in the S, N, and ORF1ab genes of the SARS-CoV-2 genome. Bacteriophage MS2 was used as an internal control to prevent false-negative results due to inhibition factors. Ct values were generated using the 7500 Software SDS 2.3 (Applied Biosystems).

Data analyses were performed by the Applied Biosystem^TM^ COVID-19 Interpretive Software, v.1.3 (Thermo Fisher Scientific, Waltham, MA, USA).

### 2.3. Whole-Genome Sequencing

Fifty-five positive samples were selected among those with a Ct value of less than 20, representing patients’ different ages, and equally distributed during in time (Table 1). The entire SARS-CoV-2 genome from 55 Sardinian viruses was obtained through NGS using Illumina (45 samples) and ONT sequencing (10 samples) platforms.

#### 2.3.1. Illumina Sequencing Method

RNA/DNA quantification was performed using a Qubit 2.0 Fluorometer (Thermo Fisher Scientific, Waltham, MA, USA) according to the manufacturer’s instructions. Libraries were prepared using the Illumina CovidSeq Test (Illumina Inc., San Diego, CA, USA). Samples were sequenced at the AMES Group, Instrumental Polydiagnostic Center Srl, Naples, Italy with an Illumina Next550 (Illumina Inc., San Diego, CA, USA), generating paired-end reads, according to the manufacturer’s instructions.

#### 2.3.2. Oxford Nanopore Technologies (ONT) Sequencing Method

Nucleic acid quantification was performed using an Epoch microplate spectrophotometer (BioTek, Winooski, VT, USA) according to the manufacturer’s instructions. Preparation of the libraries was performed at the IZS using the nCoV-2019 sequencing protocol V.3 (LoCost) [23], developed by the ARTICnetwork. Samples were sequenced at the Department of Biomedical Sciences, University of Sassari, with Flongle flow cells (R9.4.1) in a MinION platform (Oxford Nanopore Technologies). For both NGS methods, unresolved sites were completed with Sanger sequencing using the primers reported in Appendix A.

### 2.4. Bioinformatic Analysis

Illumina sequencing data processing was performed using an in-house bioinformatic pipeline.

The bcl2fastq program v2.20.0.422 was used to convert BCL files generated by the sequencing systems to standard FASTQ file formats. TrimGalore [24] was used to quality-trim the data and remove sequencing adaptors. The reads were aligned to the reference SARS-CoV-2 Wuhan-Hu-1/2019 strain (NC_045512.2) [25] using the *bwa-mem* algorithm [26]. Only reads mapping uniquely to the SARS-CoV-2 genome were retained and re-aligned using GEM [27].

Aligned bam files were both sorted and indexed with samtools [28], and then deduplicated with *Picard-tools* [29]. To obtain high-quality variants, freebayes [30] was used to call variants for each sample, using the NC_045512.2 sequence as reference genome (parameters: “ploidy1-X-u-m20—q20-F0.2”). WGSs were aligned using MAFFT 7.427 [31] and polymorphism positions were visually checked using Jalview 2.10.3B.1 [32]. Bam files generated for all virus samples were aligned against the NC_045512.2 reference genome and visually checked in IGV 2.4.14 [33]. Genome annotation was performed using the *CoVsurver* mutation app implemented in GISAID [34].

ONT sequencing data were analyzed by implementing the recommended bioinformatics pipeline developed by the ARTICnetwork [35].

Briefly, base-calling was performed using Guppy high-accuracy models (v3.5.2). As chimeric reads are the predominant source of cross-barcode assignment errors, we followed the ARTIC recommendations and demultiplexed using strict Guppy barcoder parameters to ensure that barcodes were present at each end of the fragment. The ARTIC pipeline was run using Minimap2 (v2.17) for alignment [36], and nanopolish (v0.13.2) for variant calling [37].

To overcome the 400× depth limitation of the ARTIC bioinformatics pipeline, we independently generated coverage plots using BEDtools (v2.29.2) [38] following Minimap2 alignment of the filtered FASTQ files generated using the guppyplex command.

### 2.5. Phylogenetic Analysis

The 55 newly generated Sardinian sequences were aligned against the reference SARS-CoV-2 Wuhan-Hu-1/2019 genome (NC_045512.2) [25] in BioEdit v.7.2.6.1 [39]. The evolutionary model that best fitted the data was selected by means of JmodelTest v.2.1.7 [40]. A maximum likelihood (ML) phylogeny was reconstructed in MEGA 7 [41] using the GTR + I model of nucleotide substitution. The statistical robustness of individual nodes was determined using 1000 bootstraps.

The CoVsurver mutations app of GISAID [34] was used to classify the sequences according to the different Nextstrain clades, as well as to screen Sardinian sequences for any nucleotide and amino acidic mutations in comparison with the reference genome Cov19/Wuhan/W1V04/. The lineage assessment of SARS-CoV-2 genomes was performed using PANGOLIN [42].

All newly generated sequences, along with already-published Sardinian sequences, were placed on the full phylogenetic tree of GISAID, GenBank, COG-UK and CNCB SARS-CoV-2 sequences using *Usher* [43] and run in the UCSC genome browser [44] on 7 November 2022. Individual subtrees containing 500 sequences each were extracted and visualized using Auspice [45] and NextStrain [46].

## 3. Results

### 3.1. Whole Genome Sequences Analysis

All 55 of the full genome sequences were deposited in GISAID [34] and assigned IDs from EPI_ISL_9145897 to EPI_ISL_9145951 (Table 1). NGS reads of eight whole genomes (ITA/SS-COV-1-ITA/SS-COV-8) generated full-length sequences of 29.903 bp, while the other 47 genomes presented incomplete sequences at the 5′-3′ UTRs (coverage ≥ 96.7, mean coverage 99.28%). The median depth coverage obtained for all the sequences was between 7872 and 45 (Appendix A). All genomes presented genetic variants affecting coding regions in almost all genes except NSP8, NSP11, E, M, NS6 and NS7b. Specifically, a total of 116 different protein mutations were observed (108 missense, 8 deletions and 3 premature stop codons) (Appendix A). The highest variability was observed within the Spike protein region (21.6%) and in some of the non-structural proteins (total NSP, 52.6%: NSP2, 5.2%, NSP3, 10.3%, NSP6, 5.2%, NSP12, 6%, NSP13, 7.8%), encoded by the ORF1a and ORF1b [47,48]. Amino acid mutations were also found in the N (12.9%), NS3 (6.9%), NS8 (4.3%) and NS7a (1.7%) proteins. Figure 3 reports only those mutations detected in more than 5% of samples. A further 65 mutations were found to be present in 1.8% of samples, and 21 in 3.6% of samples.

### 3.2. Phylogenetics and Molecular Epidemiology of the Outbreak

The evolutionary relationship of sequences assigned either to the first Sardinian epidemic wave (from March 2020 to August 2020, n = 5) or to the second wave (from September 2020 to January 2021, n = 50) can be observed from the reconstructed ML tree (Figure 4).

The Sardinian sequences were clustered into 5 clades and 10 lineages. The first epidemic wave was entirely represented by sequences belonging to the GR clade and the lineage B.1.1. On the other hand, sequences from the second wave were associated primarily with the GV (81.8%) and, to a less extent, with the G (3.6%), GH (1.8%) and GRY (1.8%) clades. They belonged to the B.1.177 and B.1.1 lineages along with their derived sub-lineages (Figure 4).

The full SARS-CoV-2 tree obtained from all GISAID sequences using Usher [43] and Nextstrain [46] evidenced that the 55 Sardinian SARS-CoV-2 samples are grouped into 18 different disconnected subtrees (Figure 5 and Appendix A).

Out of the 18 subtrees, 11 subtrees contain only one sequence, while 7 contain more than one sequence. The largest subtree among the latter ones (subtree 1) belongs to the clade GV, lineage B.1.177.75, and is composed of 28 out of 55 (51%) sequences, all of which can be traced back to the second wave of the Sardinian epidemic (Figure 6).

One of these sequences (COV-33) falls into a different subclade of the subtree 1 with respect to the others, most likely corresponding to a different introduction. As shown in Appendix A, the characteristic amino acid profile of this large subtree, excluding COV-33, can be summarized as NSP2-Q496H, NSP12-P323L, Spike-Y144F-A222V-D614G and N-A220V. This subtree is further divided into five subclusters differentiated by other substitutions (Appendix A). This clade also contains samples that were sequenced by other groups and deposited in GISAID; the majority of these originated from Sardinia, further confirming the importance of this clade for the second wave of the Sardinian epidemic. The closest sequences in GISAID to the most recent common ancestors (MRCAs) of the Sardinian subtrees have all been sampled from Italy. The Sardinian clade sits within a larger clade of Italian sequences (Figure 6), but contains three unique nucleotide mutations (G2293T mapped to NSP2-Q496H; A21993T mapped to Spike-Y144F; C26313T synonymous mutation in the E gene) compared to the other sequences in the larger clade.

COV-33 shares its MRCA with sequences from the Italian regions of Puglia and Campania. It belongs to the 20A.EU1 variant, which spread to Western Europe from Spain during the summer of 2020, probably through the circuit of holidaymakers starting at the end of the lockdown [49]. COV-33 shows two unique mutations for this subtree, Q836L in the Spike protein and G38stop in the N7a protein (ST4), which have no known effect on viral fitness.

The strain COV-6, collected during the beginning of November 2020, presents the deletion L140 in NSP1 and three mutations in succession: K141Q, S142V and F143I. These mutations are determined by the presence of three non-consecutive nucleotide deletions that cause the loss and re-acquisition of the reading frame. The mutation L140del was initially described in a virus isolated from Wales (EPI_ISL_473012) in May 2020. To date, this is the only Italian sequence that presents this deletion along with K141Q and S142V. The latter was firstly recorded in India (EPI_ISL_1415146) in September 2020. F143I substitution was first found in England in October 2020 (EPI_ISL_626734).

Some mutations of subtree 1 were detected for the first time in either European or Asian countries: A35E in the N protein (COV-34, 38, 39, 40, 41, 56), L186F and T26I in the NSP12 protein (COV-7 and COV-37) and T791I in the spike protein (COV-30). A97V in the NSP12 protein (COV-39) was initially found in Italy. Their biological function and references are described in ST4 along with their first detection during the SARS-CoV-2 epidemic.

In subtree 2, we found a cluster of four sequences collected in April–May 2020 during the first wave. These share their ancestral node with an isolate collected from England in April 2020 (PORT-2D1E48/2020), within a larger clade that clusters sequences from Europe (England, Germany, Slovakia, Norway and Spain), and America. COV-14 shows some characteristic mutations with respect to the node evidenced in ST4.

The third largest cluster of Sardinian samples presents a more ambiguous phylogeographic pattern (subtree 3). One of its two most basal viruses (COV-49) was sampled in Sardinia in early September 2020, whilst the other (EPI_ISL_1039469) was collected from mainland Italy during 2021. In this clade, there are also two of the Sardinian sequences generated from this study (COV-17 and COV-22), which belong to the Italian lineage B.1.177.83. The first is characterized by two mutations which were revealed for the first time in Italy (ST4). The latter presents the mutation A182S in the nucleoprotein, which was only observed in this strain when considering the whole Sardinian dataset.

From the same two basal sequences arise a cluster of strains collected from mainland Italy, as well as another containing mostly sequences from the United Kingdom, Germany and other European countries.

COV-45, COV-46 and COV-28 were clustered within a different clade composed of 45 Italian sequences generated from samples collected in Campania, Emilia Romagna, Trentino Alto Adige and Veneto (subtree 4). COV-45 and COV-46 showed additional unique mutations with respect to COV-28 (ST4).

Another cluster containing COV-19 and COV-20 sequences included strains from Germany, the United Kingdom, Denmark, Sweden, Romania and Spain, which all belong to the UK lineage 20A/S N439K variant [50,51,52,53,54,55]. Furthermore, these sequences are characterized by amino acid deletions 69 and 70, which can alter antibody recognition and are reported to affect antibody therapy treatments or immunity [56]. COV-20 also presents two further mutations with respect to this lineage (Appendix A) [57].

Eleven subtrees contained only one sequence each: COV-32, COV-54, COV-31 and COV-3 were located in different clades containing European and Sardinian viruses that presented unique and characteristic mutations (Appendix A). COV-15 and COV-25 are closely related to other European and peninsular Italian sequences. COV-53 was clustered with both Sardinian viruses and further viruses isolated from mainland Italy.

COV-9 falls into a subtree phylogenetically related to a group of Italian sequences collected in early March 2020 during the initial phase of viral propagation, mainly from the Lombardy region (EPI_ISL_542265|2020-03-01; EPI_ISL_542249|20-02-28), but also from other Italian regions. COV-44, collected during late January 2021, belongs to the English VOC Alpha and is closely related to the PHWC-4ADB88 sequence isolated in Wales and located in a subtree containing other sequences from Norway and England. COV-44 showed additional mutations with respect to the original English variant: S106del, G107del, F108del in the NSP6, L260F in the ORF1ab and P631S in the Spike. This latter mutation is typical of North American sequences. COV-4 and COV-48 clustered with isolates mainly collected from England [58].

## 4. Discussion

Our retrospective study aimed to evaluate the genetic variability of 55 newly generated SARS-Cov2 whole genome sequences collected in Northern Sardinia during the two initial epidemic waves. Five samples were collected during the first wave (April to August 2020) and fifty samples during the second wave (September 2020 to January 2021). The small sample size of the first wave reflects the low infection incidence reported in Sardinia and the inadequate diagnostic capacity at the time. We further described the transmission dynamics of SARS-CoV-2 in Sardinia, to clarify the role that Sardinia might have (or not) played in the spread of infections across the Italian peninsula.

Whole genome sequence analysis of Sardinian samples showed a significant variability among viruses circulating in this area, most probably as a result of multiple introductions of different lineages at the beginning of the second wave. Overall, we reported 108 different mutations in different protein coding regions. Two missense mutations (P323L in the Nsp12 and D614G in the Spike) were shared by all viral genomes. Further mutations appeared to be A222V in the Spike and A220V in the N protein, found in 80% of the sequences, and Q496H in the NSP2 and Y144F in the Spike, shared by about 50% of the sequences. Other mutations were also present at different levels of frequency among the sequences analyzed (Appendix A), Spike being the protein that showed the highest variability.

All the samples collected during the first wave belong to the clade GR and to the B.1.1 lineage, while during the second wave, the dominant circulating clade in Sardinia was GV, with B.1.177.75 as the dominant lineage.

The full SARS-CoV-2 tree obtained from all GISAID sequences evidenced that the Sardinian sequences generated in this study were clustered into 18 subtrees, implying at least the same number of virus introductions in Sardinia during the considered time period. It, for the most part, represents the second Italian wave sustained during Autumn 2020. The fact that 15,143 confirmed cases occurred in Sardinia during the second wave (September 2020–January 2021) and that, from our data, we estimated one introduction every 3 samples on average, might suggest that the real number of independent introductions of SARS-CoV-2 in Sardinia from elsewhere was much higher. The observation of a conspicuous number of introgressions is consistent with previous evidence, e.g., from other islands, such as Great Britain during the first wave, which recorded thousands of independent introductions [56].

Furthermore, the highly over-dispersed pattern of the subtree’s size distribution suggests that most introductions caused small or very limited outbreaks, and that only a few of them fueled the second epidemic wave on the island. The dominant cluster belongs to the B.1.177 clade that was predominant during the epidemic’s resurgence in Europe in the autumn of 2020 [59]. The wide variability in the size of Sardinian outbreaks is consistent with the known over-dispersion in SARS-CoV-2 transmission, since 80% of transmissions are generated from just 10–15% of the infected cases [60,61]. In fact, it has been recognized that most of the initial outbreaks in Wuhan [62] and the initial virus introductions in the United States have been self-limiting and disappeared without leading to large epidemics [63].

Sardinian sequences form compact clades in all but one of the seven subtrees containing more than one sequence. A parsimony approach suggests that the internal nodes of these clades correspond mostly to Sardinian viruses as well. This is consistent with locally limited transmission in Sardinia after the initial introduction, rather than movements of the virus back and forth between Sardinia and Italy (or other countries). The fact that local SARS-CoV-2 transmission drove viral spread in Autumn 2020 is not unexpected for an island where mobility to and from the Italian mainland is largely concentrated in the summer months, corresponding to the peak season for Italian summer holidays that led to a large flow of tourists.

It was often claimed in newspapers and reports that Sardinia seeded the second wave of the Italian epidemic after a rapid increase in the number of cases seen on the island during mid-August [64,65,66]. In that period, it was reported that, e.g., half of the new cases in the Lazio region had been traced to holidaymakers returning from Sardinia [67]. However, the phylogeographic data are not fully consistent with a Sardinian origin of the second wave of the Italian epidemic, as can be seen from the two largest Sardinian clusters of the second wave. As observed, the Sardinian epidemic was dominated by a large clade belonging to B.1.177.75, subtree 1, and accounting for about half of our samples. By parsimony, the ancestor of this clade likely circulated in mainland Italy. The evidence from molecular epidemiology points strongly to mainland Italy as the source of the majority of the second epidemic wave in Sardinia, rather than the other way around.

The reconstruction of the transmission history for other smaller clusters is compatible with multiple scenarios: (i) independent introductions to both mainland Italy and Sardinia; (ii) separately to Sardinia and from there to the Italian mainland, or (iii) again to mainland Italy and from there to Sardinia. This ambiguity and the fact that most basal Sardinian sequences were sampled in early September suggests that mixing in tourist hotspots on the island could be responsible for these different, but plausible, epidemiological scenarios. Otherwise, potential sampling bias, the limited number of sequences, and the difficulty in separating the phylogenetic patterns and inferring transmission directionality between Sardinia and mainland Italy during the second wave could have affected these results. Sardinia is widely assumed to have been a transmission hotspot in mid-August 2020, potentially contributing to the spread of the second Italian wave. However, the phylogeographic reconstruction of the largest clusters of Sardinian samples suggests the opposite, i.e., the Italian mainland was the source of the introduction leading to the largest component of the second epidemic wave in Sardinia. This is also consistent with the fact that lineages spreading at the end of August in Europe originated from earlier introductions at the beginning of the summer season [60]. We cannot exclude that the second Italian wave was, in part, fueled by transmission events that actually occurred in Sardinia but involved tourists from the Italian mainland, both as sources and recipients. This scenario would generate phylogenetic patterns that would not be easily separable from transmissions within mainland Italy. It is also possible that transmissions occurred mostly among tourists visiting the touristic hotspots in Sardinia during summer 2020, causing further epidemic spillovers into the local communities and then to other areas of the island.

## Figures and Tables

**Figure 1 viruses-15-00277-f001:**
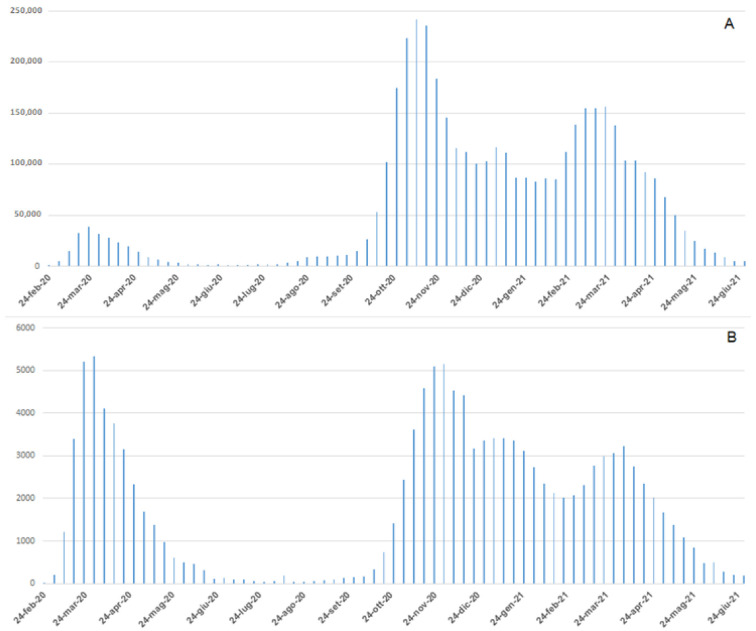
Weekly Italian trends of COVID-19 confirmed cases (**A**) and deaths (**B**).

**Figure 2 viruses-15-00277-f002:**
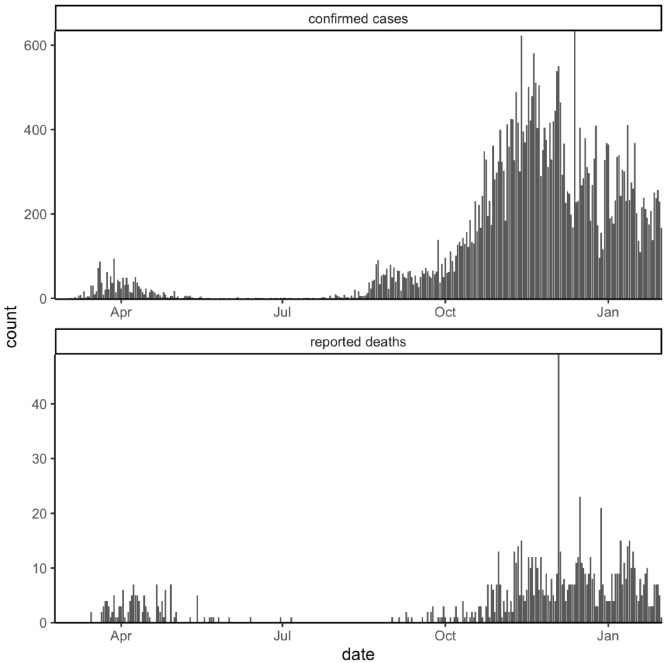
Daily Sardinian COVID-19 confirmed cases and reported deaths.

**Figure 3 viruses-15-00277-f003:**
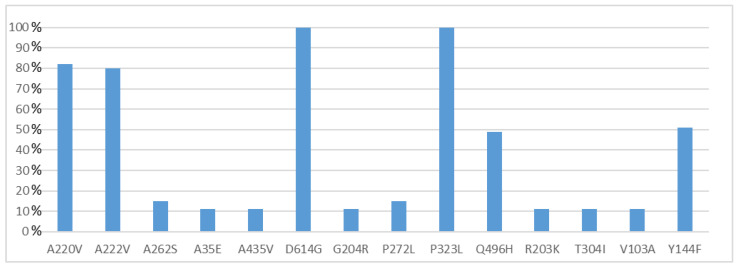
Frequency of amino acid mutations found in Sardinian samples (n = 55). Only mutations detected in more than 5% of samples were illustrated in this figure.

**Figure 4 viruses-15-00277-f004:**
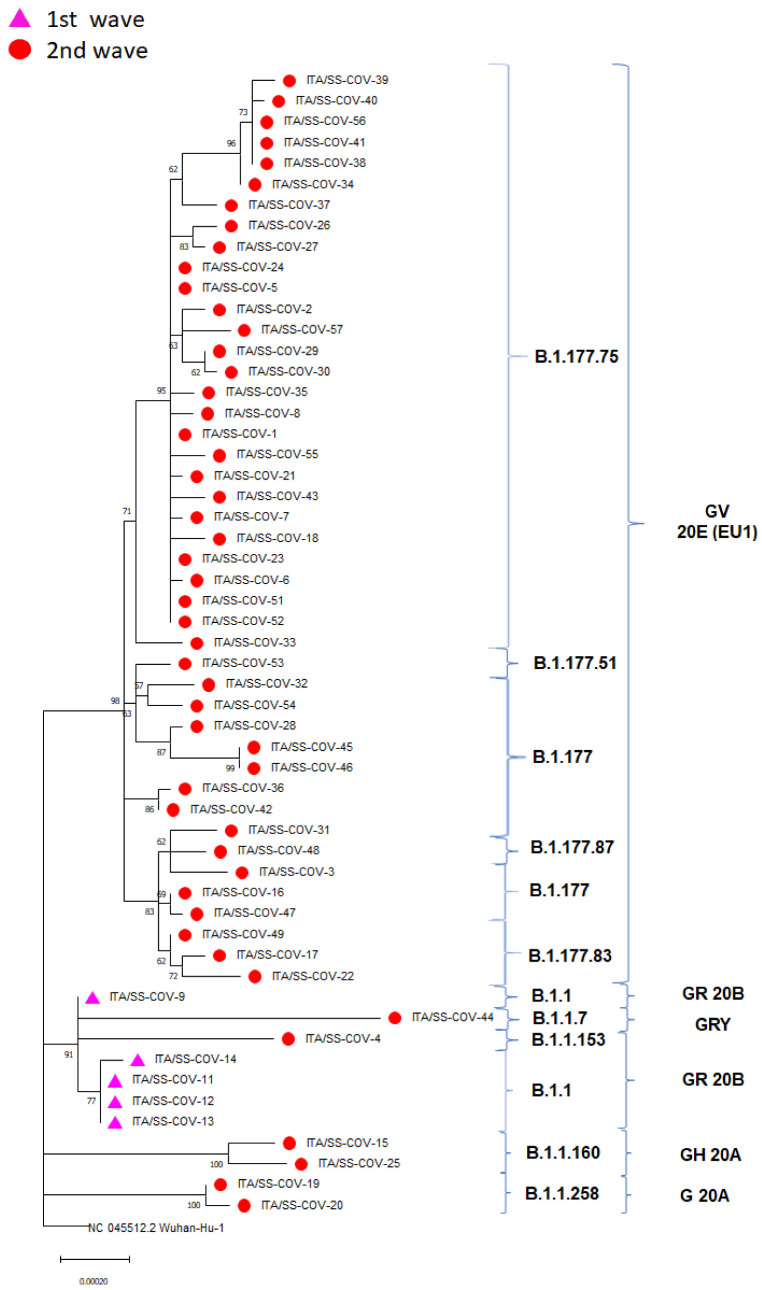
Maximum-likelihood phylogeny reconstructed using the Sardinian SARS-CoV-2 strains generated in this study. The tree is rooted to the SARS-CoV-2 reference genome Wuhan-Hu NC 045512.2. Purple triangles and red circles indicate sequences belonging to the first and second waves, respectively. SARS-CoV-2 clades and lineages are annotated using brackets.

**Figure 5 viruses-15-00277-f005:**
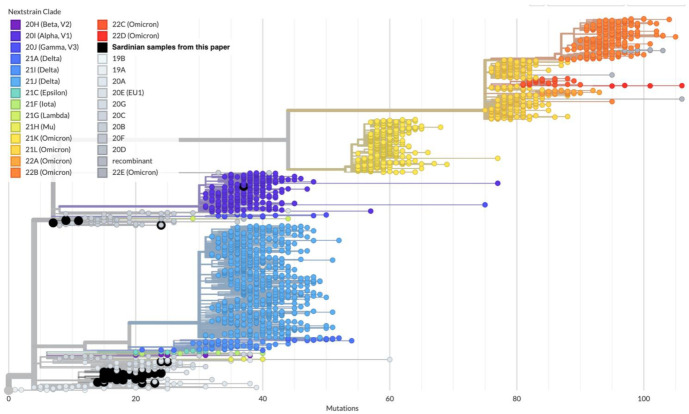
Assignment of the 55 Sardinian samples (large black circles) within the global SARS-CoV-2 phylogeny generated from Auspice/NextStrain. Most samples belong to the B.1.177 lineage and correspond to the NextStrain clade 20E (EU1).

**Figure 6 viruses-15-00277-f006:**
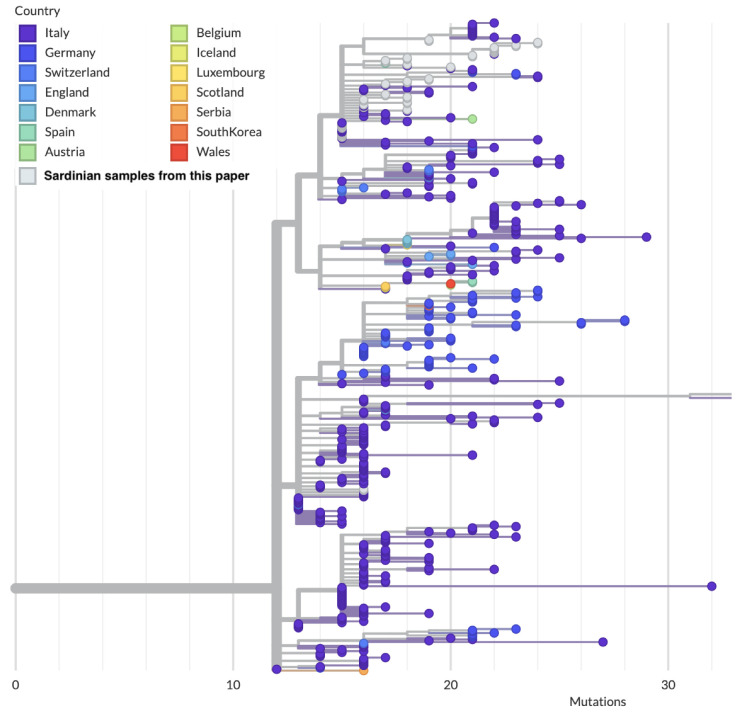
Local subtree generated from Auspice/NextStrain containing 28 out of the 55 Sardinian sequences presented in this wok (white circles). Note how most of the sequences in this subtree, including the most basal one, originate from continental Italy.

**Table 1 viruses-15-00277-t001:** Details of the Sardinian strains isolated from COVID-19-infected individuals.

Sequence ID	Age (Years)		Sex	Residence	Collection Date	GISAID Accession Number
ITA/SS-COV-21	under 12	<1	M	Ozieri	2 November 2020	EPI_ISL_9145916
ITA/SS-COV-03		10	F	Sassari	16 November 2020	EPI_ISL_9145899
ITA/SS-COV-20		10	F	Olbia	23 October 2020	EPI_ISL_9145915
ITA/SS-COV-24		12	F	Sassari	2 November 2020	EPI_ISL_9145919
ITA/SS-COV-15	13–18	13	F	Olbia	2 September 2020	EPI_ISL_9145910
ITA/SS-COV-16		13	M	Arzachena	2 September 2020	EPI_ISL_9145911
ITA/SS-COV-01		17	F	Sassari	20 October 2020	EPI_ISL_9145897
ITA/SS-COV-19		17	M	Olbia	16 October 2020	EPI_ISL_9145914
ITA/SS-COV-35		17	M	Sassari	30 November 2020	EPI_ISL_9145930
ITA/SS-COV-51		18	M	Sassari	2 November 2020	EPI_ISL_9145945
ITA/SS-COV-08	20–29	21	F	Sassari	8 October 2020	EPI_ISL_9145904
ITA/SS-COV-05		23	F	Sassari	16 October 2020	EPI_ISL_9145901
ITA/SS-COV-23		24	M	Sassari	7 October 2020	EPI_ISL_9145918
ITA/SS-COV-49		24	F	Olbia	2 September 2020	EPI_ISL_9145944
ITA/SS-COV-29		25	M	Sassari	27 November 2020	EPI_ISL_9145924
ITA/SS-COV-44		28	F	Sassari	28 January 2021	EPI_ISL_9145939
ITA/SS-COV-30		29	M	Sassari	27 November 2020	EPI_ISL_9145925
ITA/SS-COV-33	30–39	31	M	Sassari	28 November 2020	EPI_ISL_9145928
ITA/SS-COV-45		32	F	Ozieri	28 January 2021	EPI_ISL_9145940
ITA/SS-COV-31		33	F	Sassari	27 November 2020	EPI_ISL_9145926
ITA/SS-COV-25		34	M	Sassari	25 November 2020	EPI_ISL_9145920
ITA/SS-COV-27	40–49	40	M	Ozieri	27 November 2020	EPI_ISL_9145922
ITA/SS-COV-42		43	M	Sassari	22 December 2020	EPI_ISL_9145937
ITA/SS-COV-04		46	M	Olbia	2 November 2020	EPI_ISL_9145900
ITA/SS-COV-06		46	F	Sassari	4 November 2020	EPI_ISL_9145902
ITA/SS-COV-22		47	F	Olbia	17 November 2020	EPI_ISL_9145917
ITA/SS-COV-47		47	M	Sassari	2 September 2020	EPI_ISL_9145942
ITA/SS-COV-02		49	F	Sassari	2 November 2020	EPI_ISL_9145898
ITA/SS-COV-13		49	M	Sassari	15 April 2020	EPI_ISL_9145908
ITA/SS-COV-07	50–59	50	M	Sassari	12 October 2020	EPI_ISL_9145903
ITA/SS-COV-36		51	F	Sassari	1 December 2020	EPI_ISL_9145931
ITA/SS-COV-52		52	M	Sassari	2 November 2020	EPI_ISL_9145946
ITA/SS-COV-53		52	M	Olbia	25 November 2020	EPI_ISL_9145947
ITA/SS-COV-54		52	F	Sassari	3 December 2020	EPI_ISL_9145948
ITA/SS-COV-12		53	F	Sassari	15 April 2020	EPI_ISL_9145907
ITA/SS-COV-57		55	F	Sassari	3 December 2020	EPI_ISL_9145951
ITA/SS-COV-43		56	M	Sassari	25 November 2020	EPI_ISL_9145938
ITA/SS-COV-46		57	F	Ozieri	5 January 2021	EPI_ISL_9145941
ITA/SS-COV-48		58	M	Olbia	2 September 2020	EPI_ISL_9145943
ITA/SS-COV-56		59	F	Sassari	27 November 2020	EPI_ISL_9145950
ITA/SS-COV-17	60–69	63	M	Olbia	7 September 2020	EPI_ISL_9145912
ITA/SS-COV-37		65	F	Sassari	2 December 2020	EPI_ISL_9145932
ITA/SS-COV-32		68	M	Sassari	27 November 2020	EPI_ISL_9145927
ITA/SS-COV-14	70–79	72	F	Torralba	26 May 2020	EPI_ISL_9145909
ITA/SS-COV-55		73	M	Ozieri	30 December 2020	EPI_ISL_9145949
ITA/SS-COV-28		75	M	Olbia	26 November 2020	EPI_ISL_9145923
ITA/SS-COV-34	over 80	83	F	Sassari	27 November 2020	EPI_ISL_9145929
ITA/SS-COV-09		84	M	Sassari	28 April 2020	EPI_ISL_9145905
ITA/SS-COV-40		86	F	Sassari	4 December 2020	EPI_ISL_9145935
ITA/SS-COV-39		89	F	Sassari	4 December 2020	EPI_ISL_9145934
ITA/SS-COV-38		91	F	Sassari	4 December 2020	EPI_ISL_9145933
ITA/SS-COV-41		93	F	Sassari	4 December 2020	EPI_ISL_9145936
ITA/SS-COV-18		95	M	Olbia	16 October 2020	EPI_ISL_9145913
ITA/SS-COV-11		99	F	Torralba	6 May 2020	EPI_ISL_9145906
ITA/SS-COV-26		100	M	Ozieri	27 November 2020	EPI_ISL_9145921

## Data Availability

Data are contained within the article and Appendix A.

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
