# Peer review of "Origin, Genetic Variation and Molecular Epidemiology of SARS-CoV-2 Strains Circulating in Sardinia (Italy) during the First and Second COVID-19 Epidemic Waves"

_viruses, 2023, doi:10.3390/v15020277_

Round 1
Reviewer 1 Report
The results are clearly presented and in a logical order.
- It would be good to include a couple of sentences on the limitations of your study. For example, you already mention the difficulty in separating the phylogenetic patterns and infer transmission directionality between Sardinia and mainland Italy during the second wave. It would be good if you were additionally able to discuss potential sampling bias (as you are working on a limited number of sequences).
- It would be good to indicate if the tourist season during the second wave was at a similar capacity to previous years (as this would indicate the scale of transient population), and the relative % of tourists from the Italian mainland (if available) as that would link with your discussion.
- How similar are these results to previous EID outbreaks in Sardinia? For example, are prior influenza outbreak patterns comparable to the first wave (as they both took place during winter periods)?
- Please add 1-2 sentences on the public health measures that were introduced in Sardinia as a result of the second wave.
Author Response
The results are clearly presented and in a logical order.
- It would be good to include a couple of sentences on the limitations of your study. For example, you already mention the difficulty in separating the phylogenetic patterns and infer transmission directionality between Sardinia and mainland Italy during the second wave. It would be good if you were additionally able to discuss potential sampling bias (as you are working on a limited number of sequences).
R: dear review, thank for this important suggestions. This and other study limits have been highlight in discussion following your ideas and considerations.
- It would be good to indicate if the tourist season during the second wave was at a similar capacity to previous years (as this would indicate the scale of transient population), and the relative % of tourists from the Italian mainland (if available) as that would link with your discussion.
R: unfortunately no official data about the movements from Italian mainland and Sardinian island are available.
- How similar are these results to previous EID outbreaks in Sardinia? For example, are prior influenza outbreak patterns comparable to the first wave (as they both took place during winter periods)?
Dear review, considering the data published by the Italian integrated surveillance system of seasonal influenza, the prior influenza took place every year during November-February that could be defined as winter period in Italy. Otherwise, the first wave of COVID-19 in Italy took place in March-May. These two periods are not comparable considering the completely different risk factors (i.e., temperature, humidity, wind, human behaviour). Otherwise, the second COVID-19 wave occurred during the winter period reporting many more cases respect to those reported during all the seasonal influenza waves. Given your important suggestion we agree with you in the need of discuss this point and we included some note in introduction and discussion. Thank you very much.
- Please add 1-2 sentences on the public health measures that were introduced in Sardinia as a result of the second wave.
R: some information and the reference have been included in introduction following this suggestion

Reviewer 2 Report
dear authors,
please provide in the introduction structure of SARS-CoV-2 in relation to other genomic strains
in discussion please discuss how genetic variability affects the pathogenesis
give limitations of the study
gives sound conclusion
i suggest the following references
Moubarak M, Kasozi KI, Hetta HF, Shaheen HM, Rauf A, Al-Kuraishy HM, Qusti S, Alshammari EM, Ayikobua ET, Ssempijja F, Afodun AM. The rise of SARS-CoV-2 variants and the role of convalescent plasma therapy for management of infections. Life. 2021 Jul 23;11(8):734.
Alomair BM, Al-Kuraishy HM, Al-Buhadily AK, Al-Gareeb AI, De Waard M, Elekhnawy E, Baiha GE. Is sitagliptin effective for SARS-CoV-2 infection: false or true prophecy?. Inflammopharmacology. 2022 Sep 30:1-5.
Alomair BM, Al-Kuraishy HM, Al-Buhadily AK, Al-Gareeb AI, De Waard M, Elekhnawy E, Batiha GE. Is sitagliptin effective for SARS-CoV-2 infection: false or true prophecy?. Inflammopharmacology. 2022 Sep 30:1-5.
Batiha GE, Shaheen HM, Al-Kuraishy HM, Teibo JO, Akinfe OA, Al-Garbee AI, Teibo TK, Kabrah SM. Possible mechanistic insights into iron homeostasis role of the action of 4-aminoquinolines (chloroquine/hydroxychloroquine) on COVID-19 (SARS-CoV-2) infection. Eur Rev Med Pharmacol Sci. 2021 Dec 1;25(23):7565-84.
Author Response
dear authors,
- please provide in the introduction structure of SARS-CoV-2 in relation to other genomic strains
R: we have improved the introduction accordingly. In the text a reference suggested by the referee was also cited.
- in discussion please discuss how genetic variability affects the pathogenesis
R: the main aim of this work was to assess genetic variability of Sardinian Sars-Cov-2 strains in order to evaluate possible epidemiological scenarios regarding the first and second epidemic waves. Unfortunately, as stated in the materials and methods section, no clinical information was available for the infected patients so we are not able to correlate the clinical data with the Sars-Cov-2 variants analysed in this study.
- give limitations of the study
R: dear review, thank for this important suggestions. This and other study limits have been highlight in discussion following your ideas and considerations.
- gives sound conclusion
R: in the last part of the discussion section we have made some hypothesis about the viral spread between the insular and peninsular areas of Italy that in our opinion constitute an appropriate interpretation of our data.
- I suggest the following references
R: We thank the referee for the suggestion that we complied accordingly
